# Triboelectric Charging Properties of the Functional Groups of Common Pharmaceutical Materials Using Density Functional Theory Calculations

**DOI:** 10.3390/pharmaceutics16030433

**Published:** 2024-03-21

**Authors:** James R. Middleton, Mojtaba Ghadiri, Andrew J. Scott

**Affiliations:** School of Chemical and Process Engineering, University of Leeds, Leeds LS2 9JT, UK; m.ghadiri@leeds.ac.uk (M.G.); a.j.scott@leeds.ac.uk (A.J.S.)

**Keywords:** triboelectric charging, contact electrification, paracetamol, density functional theory, DFT, first principles, ab initio

## Abstract

Triboelectrification is a ubiquitous and poorly understood phenomenon in powder processing, particularly for pharmaceutical powders. Charged particles can adhere to vessel walls, causing sheeting; they can also cause agglomeration, threatening the stability of powder formulations, and in extreme cases electrostatic discharges, which present a serious fire and explosion hazard. Triboelectrification is highly sensitive to environmental and material conditions, which makes it very difficult to compare experimental results from different publications. In this work, density functional theory (DFT) is used to investigate the charge transfer characteristics of several functional groups of paracetamol in order to better understand the mechanisms of charging at the nanoscale and the influence of the environmental and material properties on charge transfer. This is achieved by studying the structure and electronic properties at the molecule–substrate interface. Using this molecule–substrate approach, the charging contributions of individual functional groups are explored by examining the Hirschfeld charges, the charge density difference between the molecule and substrate, the density of states, and the location of the frontier orbitals (HOMO and LUMO) of a paracetamol molecule. Charge density difference calculations indicate a significant transfer of charge from the molecule to the surface. Observable regions of electron density enrichment and depletion are evident around the electron-donating and -withdrawing groups, respectively. The density of states for the paracetamol molecule evolves as it approaches the surface, and the band gap disappears upon contact with the substrate. Hirshfeld charge analysis reveals asymmetry in the charge redistribution around the molecule, highlighting the varying charging tendencies of different atoms.

## 1. Introduction

Triboelectrification is a persistent problem in the pharmaceutical industry and in powder processing more generally. For the case of powders and grains, it is caused by the transfer of charged species during physical contact which, after separation, leave a residual charge on the particles [1]. Due to electrostatic forces resulting from triboelectric charging, charged particles can cause significant processing issues such as creating blockages at inlets, adhering to vessel walls and causing a phenomenon known as sheeting [2], or producing electrostatic discharges, which present a significant fire and explosion hazard [3]. These issues are highly pronounced in the pharmaceutical industry, where triboelectric charging can destabilise powder formulations, threatening blend homogeneity [4], which can cause batches to miss regulatory requirements. It can also severely impact the efficacy of aerosolised drug delivery systems, such as dry powder inhalers [5]. Triboelectrification is a poorly understood phenomenon [6]. However, with recent advances in experimental [7,8] and computational [9,10] approaches, it is hoped that the phenomenon can be better understood to mitigate these negative effects.

Modelling triboelectric charging using first-principles calculations is an active area of research, with the majority of recent publications focused on improving the design and performance of triboelectric nanogenerators. However, the accurate ab initio prediction of triboelectric charging could offer several significant benefits to the pharmaceutical industry. Digital drug design has emerged as a promising technique in the area of drug discovery and development, which leverages computational methods to significantly improve the way we search for new pharmaceutical agents [11]. While digital drug design and computational methods have made significant strides in various aspects of drug discovery and development, computational methods can also be applied to better understand and improve the manufacturing processes of pharmaceutical materials [12]. However, predicting the tendency of a drug to become triboelectrically charged during transport remains a challenging task.

In the realm of drug development, precise predictions of triboelectric charging tendencies could lead to improved drug formulation and stability, enabling the a priori identification of potential issues caused by agglomeration, adhesion, or electrostatic discharges during transport and processing [13]. The impact of surface functional groups on triboelectric charging tendency is well known in various fields [14,15,16]. By considering the impact of functional groups on triboelectric charging tendencies, drug designers can make informed choices about the composition and structure of drug molecules. Careful selection and placement of functional groups might help minimize undesirable triboelectric charging effects while maintaining drug efficacy. This is especially relevant for aerosolised drug particles, where triboelectric charging can significantly impact pulmonary drug delivery.

Ab initio calculations can be used to calculate the electronic structures of individual molecules and even analyse the electronic properties of individual functional groups within those molecules. This enables researchers to gain valuable insights into the behaviour of small systems that may not be feasible to study directly in experiments. In the context of organic electronic devices, Li et al. [17] investigated how polymer surface treatments impact surface electronic structure by putting a single monomer on a NiO surface. This work inspired the work of Wu et al. [18], who used this same molecule–substrate approach to assess the properties of different polymers for their use in triboelectric nanogenerators. More recently, Nan et al. [9] used DFT to study the charge transfer of several polymers between an idealised water layer to comprehensively study the contact electrification mechanism at water polymer interfaces.

In our recent work, DFT calculations have been used to predict variations in the effective work function of different surfaces of pharmaceutical crystals, which is a key quantity in the modelling for triboelectric charge transfer [19]. However, research on the application of DFT to predict the triboelectric charging tendency of pharmaceutically relevant materials is sparse [20]. Nonetheless, DFT is used extensively to model the electronic structure of a wide range of materials [21], including pharmaceuticals [22], providing a strong methodological basis for this work.

We analyse the charging tendency of the functional groups of paracetamol using a molecule–substrate approach analogous to other works [18]. Molecules of paracetamol are placed on two atom-thick surfaces of several metals that have a range of work functions. Looking at the charge transfer of each functional group under a variety of conditions using Hirshfeld charge analysis, the charge density difference between the surface and the paracetamol molecule is quantified, and the physical location of frontier orbitals (HOMO and LUMO) is determined. These properties are quantified as a function of distance from the surface. The objective of this work is to provide an insight on how the functional groups exposed at the facets of pharmaceutical crystals might influence triboelectric charging.

## 2. Theoretical Approach

A molecule–substrate approach [17,18] was used to examine the electronic structure properties of paracetamol when in contact with metal surfaces in order to gain insight into the triboelectric charge transfer properties of the molecule. This procedure involves placing a molecule on the surface of a material and observing how contact with that surface influences its electronic structure. This procedure has been used to study the triboelectric properties of monomers of polymeric materials by Wu et al. [18] and Nan et al. [9]. Interestingly, a similar approach is often applied to study corrosion inhibition, which has been studied extensively using DFT [23]. The procedure is as follows: the molecule of interest is placed on a slab of material. A geometry optimisation is performed to relax the structure and place it in the most energetically favourable position. This is then followed by a single-point energy DFT calculation to determine the electronic structure’s properties.

In this work, a combined DFT and molecular mechanics (MM) approach was used to simulate the electronic structure and relax the geometry of the systems. In a recent work by Marchese-Robinson et al. [24], it was shown that the COMPASSIII force field [25] gave excellent results even compared with the computationally expensive DFT-based approach. MM is much faster, requires much less computational resources, and gives comparable results to DFT methods. Therefore, it was employed in this work to perform the geometry optimisation calculations, followed by DFT calculations to calculate the electronic structure.

Materials Studio 2023 software was leveraged to construct the periodic cells used in this work. MM calculations were performed using the Materials Studio Forcite Molecular Mechanics Tool. MM was used to perform geometry optimisation calculations, applying a COMPASSIII force field throughout. With force field-assigned charges, Ewald and atom-based energy summation methods were used for the electrostatic and van der Waals terms, respectively. The convergence criteria were an energy tolerance of 5 × 10^−6^ kcal/mol/Å, a force tolerance of 0.1 kcal/mol/Å, stress tolerance of 0.02 GPa, and a displacement tolerance of 5 × 10^−4^ Å. Electronic structure calculations were performed using the CASTEP density functional theory code [26]. For all calculations in this work, the Perdew–Burke–Ernzerhof [27] exchange correlation functional with the Tkatchenko–Scheffler (TS) dispersion correction scheme [28] was used unless stated otherwise. A minimum kinetic energy cutoff of 700 eV was used throughout. A Monkhorst–Pack grid of 2 × 2 × 1 is used for energy calculations, and 4 × 4 × 2 for density of states (DOS) calculations. For the electronic minimisation of the total energy per atom, a convergence tolerance of 5.0 × 10^−7^ eV was imposed. To ensure the validity of the combined MM and DFT approach, the performance of the COMPASSIII was tested against DFT-based approaches and the experimental results of Wyckoff [29] and Bruhn [30]. All geometry optimisation and DFT calculations were performed at 0 K. DFT calculation parameters for this test are shown in Table 1. Visualizations of each unit cell used in this test are shown in Figure 1.

The lattice parameters predicted by simulations were shown to have excellent agreement with experimental data in all cases (Table 2). However, using a MM-COMPASSIII-based approach offers significant reductions in computational resources and simulation time. For this reason, COMPASSIII was used for all geometry optimisation calculations in this work. Density of states (DOS) calculations were performed, which typically require more precise settings, to ensure well-converged results. A kinetic energy cutoff of 700 eV was used throughout this work. For energy calculations, a Monkhorst–Pack k-point sampling of 2 × 2 × 1 was used, and for DOS a sampling of 4 × 4 × 1 was used.

In an attempt to quantify the triboelectric charging characteristics of different atoms of a molecule, Hirshfeld charge analysis [31] was used to quantify the charge transfer between atoms and different surfaces. Charge density difference calculations were also performed to investigate the regions of the molecule which exchange charge density with the surface. In the work of Wu et al. [18], a molecule was placed on a metal slab several atoms thick; however, in the work of Nan et al. [9], molecules were placed on only a single layer of ordered water. Here, we present the convergence of Hirshfeld charges with the layer thickness, as shown in Figure 2. Therefore, a slab thickness of two atomic layers was deemed to be sufficiently converged for these calculations and was used throughout this work.

To gain insight into potential charge transfer, charge density changes were calculated to show the restructuring of charge when paracetamol is placed on a metal surface. The charge density difference induced by the molecules placed on the slab is given by
(1)$$\Delta \rho =\rho_{surf-mol}-\rho_{surf}-\rho_{mol}$$

The charge density of the surface ($\rho_{surf}$) and molecule ($\rho_{mol}$) were subtracted from the charge density of the combined surface and molecule ($\rho_{surf-mol}$), thereby enabling the visualisation of areas of electron depletion and electron enrichment due to contact. This requires three separate calculations involving the isolated molecule, the isolated surface, and the whole system. Calculations were performed at different distances from the surface, where a molecule was placed on a fixed surface and the structure of the molecule was then relaxed to obtain the first system at position ($d_{0}$); the conformation of the molecule was then fixed in order to keep the distance of the functional groups of the molecule and the surface consistent, and then incrementally moved away from the surface into positions ($d_{0}+0.08\;Å$, $d_{0}+0.16\;Å$, …). In this way, the influence of proximity to the surface and the electronic structure was investigated.

In this work, the interaction of a single molecule of paracetamol with several metal surfaces was investigated. Two atomic layer surfaces of (111) aluminium, copper, and nickel were selected based on their common FCC structure and their distinct work functions, which historically have often been correlated with charging propensity [32]. Work function and Fermi energy were analysed to determine the driving force for charge transfer. Charge density difference calculations were performed to investigate how electron density shifts around functional groups due to contact. The DOS was investigated, looking at the frontier orbitals and band gap. Finally, the Hirshfeld charges of the molecule were calculated to investigate how atomic charges around the molecule might shift due to contact.

## 3. Results and Discussion

The variations of the electrostatic potential of a periodic cell of paracetamol in the absence, and presence, of the aluminium (111) surface are shown in Figure 3. The electrostatic potential of the whole system aligns well with the electrostatic potential of the isolated components as expected. There is a noticeable difference in the vacuum energy of the isolated paracetamol compared to the vacuum energy of the isolated metal surface or the whole system; this is expected to create a dipole moment between the surfaces, and therefore some charge redistribution takes place at the surface. The electrostatic peaks of the surface-molecule system are slightly depressed compared to the isolated systems, which could influence the electrostatic barrier to charge transfer [33]. This observation was consistent across all surfaces tested; additional plots are available in the Appendix A.

To examine any potential driving force for charge transfer, the work function (and effective work function in the case of paracetamol) was calculated based on Equation (2).
(2)$$\varphi =E_{vac}-E_{F}$$
where φ is the work function, $E_{vac}$ is the vacuum energy, and $E_{F}$ is the Fermi energy of the system. The calculated values are displayed in Table 3. However, these theoretical work functions are not expected to correlate well with experimental work functions of real materials, as the theoretical calculations have been found to depend strongly on slab thickness [34]. Nevertheless, they are a useful indicator of the direction of charge transfer in this system. In a study conducted by Nan et al. [9], it was observed that when employing DFT for model charge transfer, a substrate with a thickness of just a single atomic layer proved adequate to capture the charge transfer behaviour. This observation aligns closely with the results obtained here, as illustrated in Figure 2. Consequently, we adopted a restricted slab thickness in our computations to minimize unnecessary computational expenses. Based on the work functions reported here, it is predicted that the paracetamol molecule charges negatively against all the metal surfaces modelled here, with nickel charging most strongly.

The Fermi energy values of each system investigated show a similar overall trend but are not shown for brevity. It becomes less negative in all cases as the paracetamol surface approaches the metal surface, with the rate of change becoming greater as the molecules approach the surface. A shift in the Fermi level implies there is a shift in the occupied states of a material, therefore implying charge transfer. There is a difference in magnitude of the Fermi level shift, which follows the order of the work function, i.e., Al > Cu > Ni, with Fermi level shifts of 0.188 eV, 0.249 eV, and 0.260 eV, respectively. This order of work functions aligns with the experimental data reported by Kawano et al. [35]. Therefore, the difference in Fermi level shift could be correlated with the magnitude of charge transfer. Plots illustrating the relationship between the Fermi level and the distance from the surface are provided in the Appendix A.

Charge density difference plots are commonly used when applying DFT to study triboelectric charging [18,36,37]. Typically, they are employed to examine the redistribution of charges in molecules that are chemically bonded to a surface. In triboelectric charge transfer studies, these plots are utilized to highlight charge transfer occurring between surfaces. Figure 4 and Figure 5 show the charge density difference iso-surface due to the contact of the paracetamol molecule with the metal surface. Yellow iso-surfaces indicate areas of electron depletion (positive charging) and blue areas indicate areas of electron enrichment (negative changing). These surface regions were set to have a threshold value of 7 × 10^−3^ electron/Å^3^, i.e., shifts in electron density below this value are not displayed by the surface, and only the regions that exceed this value are visible. The observed regions of electron density difference align well with initial expectations. The region below the electron withdrawing carbonyl group consistently shows a large region of electron depletion at the surface, in all systems tested.

The region below the electron-donating hydroxyl group shows a region of electron enrichment in all systems. Surprisingly, there was a clear region of enrichment around the methyl group in all systems tested. When examining charge density differences on various surfaces, it is evident that less charge redistribution occurs on the aluminium surface compared to the other surfaces (Figure 4). The visible redistribution on the surface is likely due to the charge density difference of the surface being too small to meet a threshold value, which implies a lower magnitude for charging of the surface. There are regions of charge density difference spread across the surface atoms of nickel and copper, showing more concentrated regions of charge density difference around the surface atoms. An interesting difference between the surfaces tested is the region below the phenyl group on copper, which shows a region of electron enrichment, whereas the regions below nickel and aluminium appear to charge more positively, showing subtly in the expected charging properties of other functional groups. The regions of charge density difference grow as the molecule approaches the surface; however, interestingly, the electron depletion effect of carbonyl group occurs before the enrichment effect of hydroxyl, which is either due to a longer range of action or a greater magnitude of charge transfer. Additional plots showing the side profile view of charge density differences in the iso-surfaces of copper and nickel are available in the Appendix A.

Figure 6, Figure 7 and Figure 8 display the 1D planar averaged charge density difference, a more quantitative measure of charge density difference, showing the net charge redistribution between the surface and the molecule. In all the surfaces tested, there appears to be a sharp decrease in the electron density in the region surrounding the molecule, which predicts a net electron transfer to the surface and a positive charging of the paracetamol molecule. Additionally, the strongest shift in charge density difference from the molecule is observed when the paracetamol molecule is placed on the nickel surface, followed by copper and then aluminium. This matches with the order of work functions presented in Table 3. Each surface has a unique fingerprint of the charge restructuring in the region surrounding the actual surface. There is a clear evolution in charge density difference as the molecule approaches the surface. There is little to no charge restructuring 1.25 Å from the optimised position on the surface.

The frontier orbitals of the system are reported to be of crucial importance to charge transfer [38,39,40]. These orbitals contain electrons of relatively low energy, and therefore in a low-energy process like triboelectric charging, these electrons are the ones that are most likely to move between molecules. Figure 9, Figure 10, Figure 11, Figure 12 and Figure 13 show the DOS of paracetamol molecules at various positions in these systems. Interestingly, a clear band gap is observable in these systems, where the paracetamol molecule is placed at a sufficient distance from the surface, as expected for insulating materials. However, as the DOS evolves when the molecule is moved closer to the surface, the electron states near the Fermi level shift appear to broaden, so that the band gap disappears. The same effect was observed by Yoshida et al. [41], where the calculated bad gap was observed to disappear after charge transfer, implying charge transfer is occurring. These results also have implications for the electron transfer mechanism for the overlapping electron cloud, as described by Lin et al. [42], which supports this finding. This shift is present in all surfaces tested. The pattern of the shift is also unique depending on the surface that the paracetamol molecule is placed on.

In Figure 14, Figure 15 and Figure 16, the Hirshfeld charges are visually represented on their corresponding atoms within the systems. No significant shift in Hirshfeld charges is observable on the second layer atoms of the surface, which predicts that the majority of the charge transfer occurs from the top layer of the material (Figure 14). As expected, when the molecules are 5 Å away from the surface, there is no charge deformation, and the surface is neutral in all systems tested. Interestingly, when contacted with paracetamol, the Hirshfeld charges on the surface align reasonably well with the charge density difference of the iso-surfaces. A region of negative Hirshfeld charge (electron enrichment) is present in the area surrounding the hydroxyl group. Likewise, a region of positive Hirshfeld charge (electron depletion) is present around the carbonyl group. Unusually, the charge density difference of the iso-surface and the surface Hirshfeld charge shift at the centre of the phenyl ring do not predict the same shift in charge density. Hirshfeld charges indicate there is a region of positive Hirshfeld charge (electron depletion) at the surfaces of aluminium and copper, but for nickel it is unchanged. However, this behaviour of nickel and the phenyl ring correlates well with the predicted location of the LUMO shown in Figure 9. There appears to be different magnitudes of charging for different functional groups, with the carbonyl group charging most strongly with copper and weakest with nickel, whereas the hydroxyl group charges most strongly with nickel and weakest with copper (Figure 15).

The Hirshfeld charges of the isolated molecules are as expected, with the positive Hirshfeld around the hydrogen and negative around the more electronegative nitrogen and oxygen atoms (Figure 16 and Figure 17). For paracetamol, when contact was made with the surface, the same general trend in Hirshfeld charge was observed; however, a noticeable shift in the Hirshfeld charges was observed depending on atom type and contacting surface (Figure 18 and Figure 19). Significant deviations arise when comparing the atomic charges of the surface and the carbonyl group. The carbonyl group consistently loses electron density when in contact with the surface, posing challenges in comprehending the direction of the charge transfer.

## 4. Conclusions

In this work, the interactions between paracetamol and metallic surfaces of aluminium, copper, and nickel have been studied using DFT to try an elucidate the mechanisms of triboelectric charging using a molecule–substrate approach. Analysing the characteristics of a single molecule of paracetamol reveals its asymmetric charging tendency. This is observable from the charge density difference plots and the Hirshfeld charge analysis, which shows regions of electron enrichment around the electron-donating hydroxyl group, and regions of electron depletion around the electron-withdrawing carbonyl group. The 1D planar averaged charge density difference predicts a charge density depletion from the molecule, implying charge transfer to the surface. This is further supported by the shift in the Fermi energy level, as the molecule approaches the surface with implied charge transfer. The DOS of the paracetamol molecule evolves as the paracetamol approaches the surface. The band gap is found to disappear when paracetamol is in ‘contact’ with the surface, which implies that there is now an overlap between the valence and conduction bands, and there is now free movement of electrons between the surface and the molecule. This effect was present in all the systems analysed. Additionally, each surface is shown to have a unique effect on the frontier orbitals of paracetamol, which suggests complex charging behaviour. The Hirshfeld charge analysis on the movement of charge gives mixed results. The shift in Hirshfeld charges at the surface of the metals align well with the regions of electron enrichment and electron depletion above the hydroxyl group and carbonyl group, respectively. Interestingly, the shift in atomic charges is different depending on the metal surface contacted. However, there is poor agreement of the charging propensity of the central phenyl ring; the results of the Hirshfeld charge analysis deviate strongly from the regions of electron enrichment and depletion predicted by the charge density difference of iso-surfaces.

In conclusion, the first-principles calculations show that, as expected, the functional groups exhibit different charging tendencies, as given by charge density difference and Hirshfeld charge analysis. Some charge transfer is predicted by the Fermi level shift and shift in electron density towards the surface. DOS calculations show the disappearance of the band gap towards the surface. It is postulated that this is due to the overlapping of electron orbitals, responsible for triboelectric charge transfer, providing supporting evidence for a universal electron transfer mechanism.

## Figures and Tables

**Figure 1 pharmaceutics-16-00433-f001:**
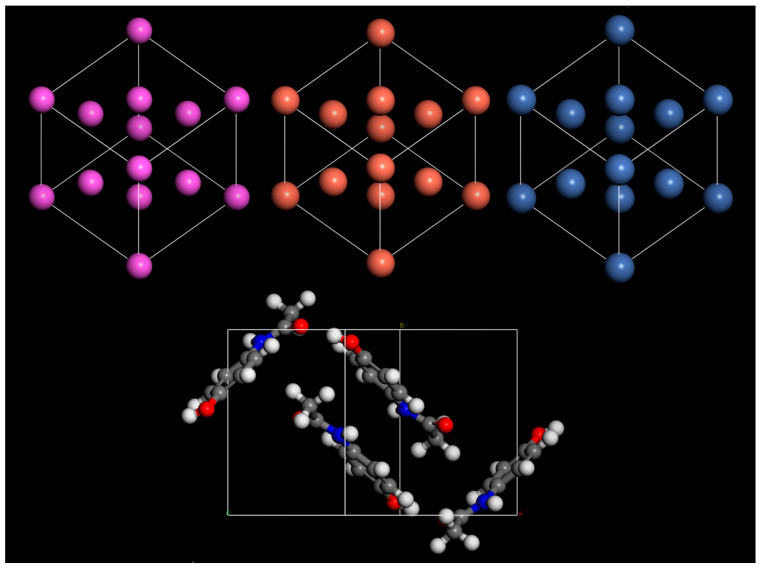
Bulk unit cell of aluminium, copper, nickel, and paracetamol.

**Figure 2 pharmaceutics-16-00433-f002:**
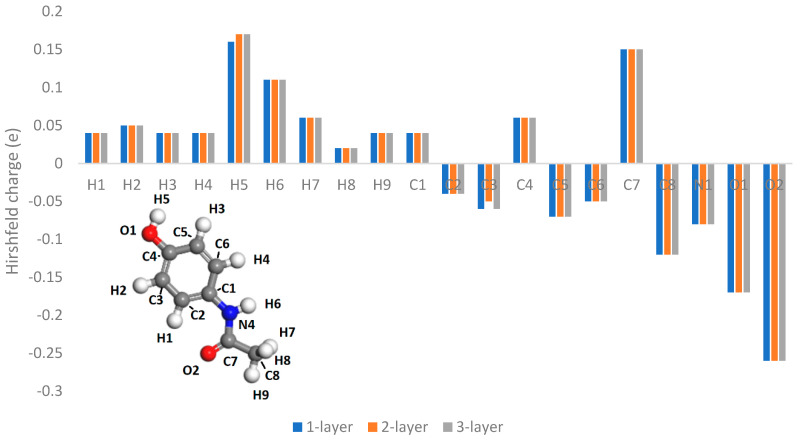
Sensitivity analysis of Hirshfeld charge versus number of atomic surface layers, carried out on an aluminium (100) surface, using a 600 eV kinetic energy cutoff, and 2 × 2 × 1 Monkhorst–Pack grid.

**Figure 3 pharmaceutics-16-00433-f003:**
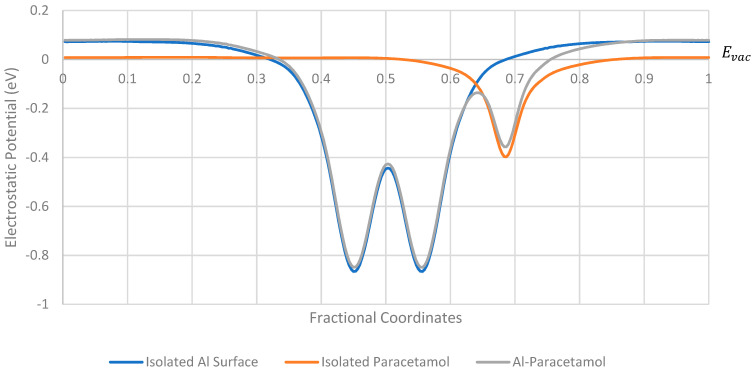
Electrostatic potential of aluminium (111) surface with a single paracetamol placed on its surface at its structurally optimised position. Electrostatic potential of the isolated surface and isolated molecule also shown.

**Figure 4 pharmaceutics-16-00433-f004:**
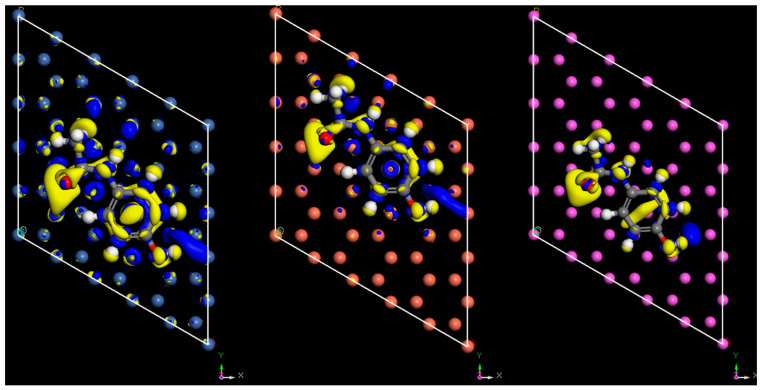
Charge density difference iso-surface of paracetamol on nickel (**left**), copper (**middle**), and aluminium (**right**) at the geometry-optimised position from the surface, $d_{0}$. Top view. Iso value = 7 × 10^−3^ electron/Å^3^.

**Figure 5 pharmaceutics-16-00433-f005:**
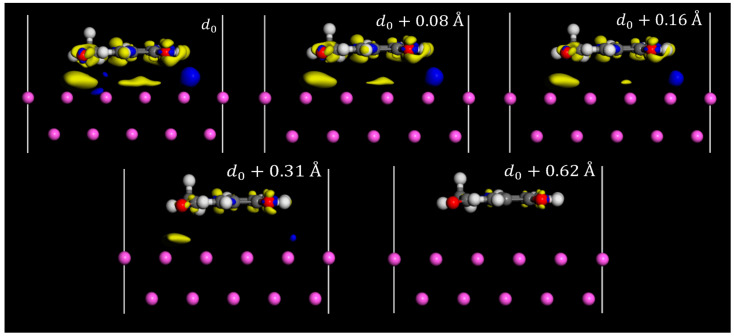
Charge density difference iso-surface of paracetamol on aluminium (111) surface at several distances. Side view. The geometry-optimised position of paracetamol is indicated by $d_{0}$. Iso value = 7 × 10^−3^ electron/Å^3^.

**Figure 6 pharmaceutics-16-00433-f006:**
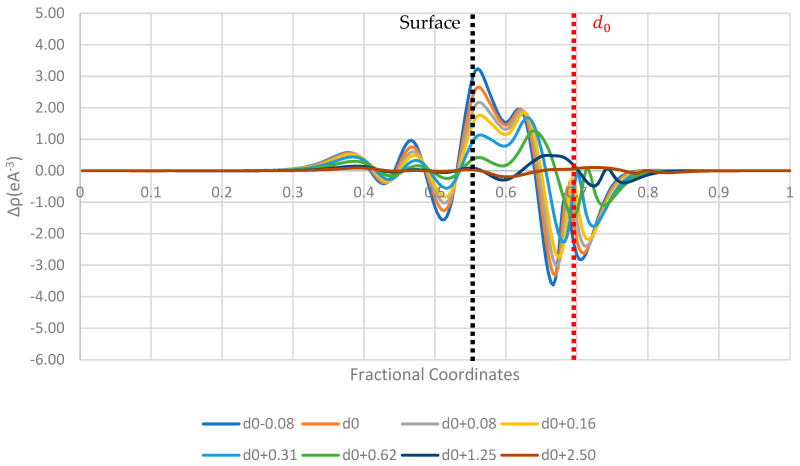
Planar-averaged charge density difference of the aluminium–paracetamol system at several distances.

**Figure 7 pharmaceutics-16-00433-f007:**
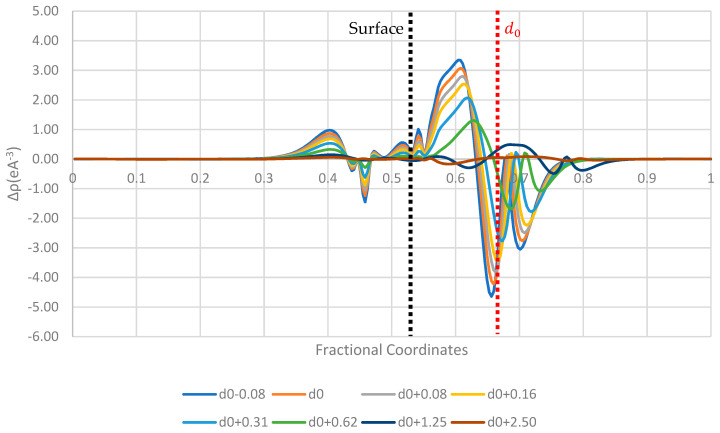
Planar-averaged charge density difference of the copper–paracetamol system at several distances.

**Figure 8 pharmaceutics-16-00433-f008:**
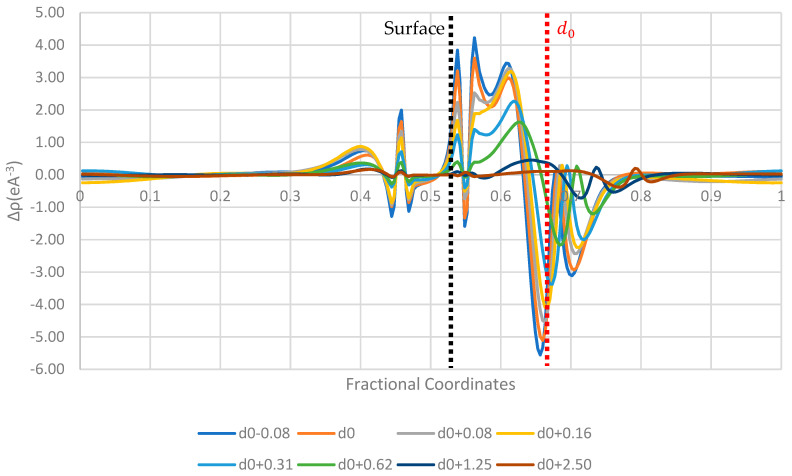
Planar-averaged charge density difference of the nickel–paracetamol system at several distances.

**Figure 9 pharmaceutics-16-00433-f009:**
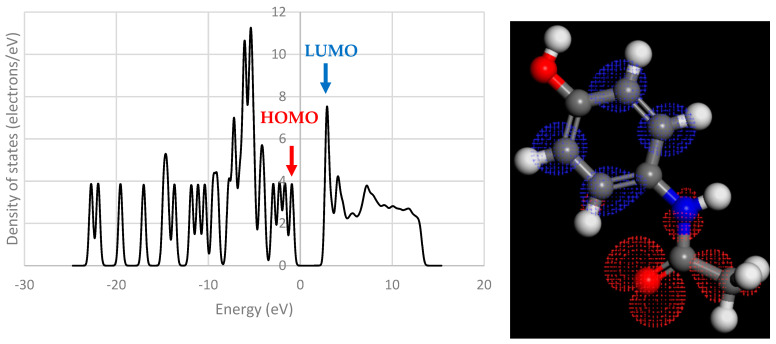
DOS of a paracetamol 5 Å away from its optimised position on an aluminium (111) surface (**left**). Iso-surface of HOMO (red) and LUMO (blue) orbitals obtained from DOS calculation on molecule (**right**).

**Figure 10 pharmaceutics-16-00433-f010:**
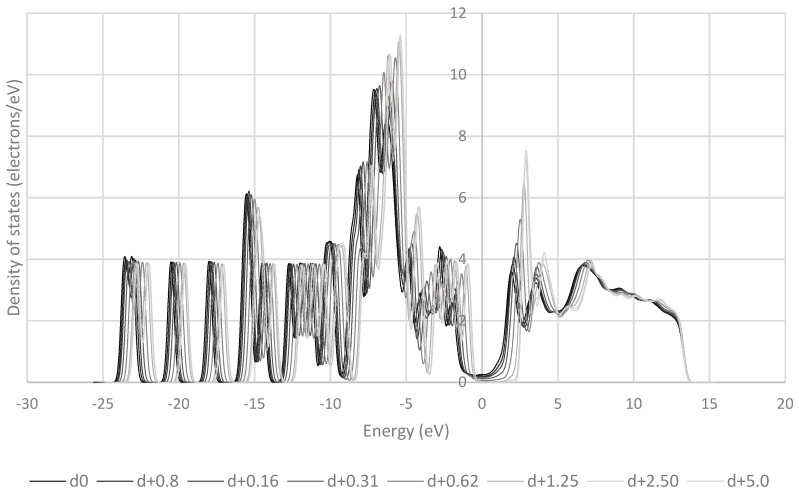
Evolution of the DOS of paracetamol as the molecule is withdrawn from an aluminium (111) surface.

**Figure 11 pharmaceutics-16-00433-f011:**
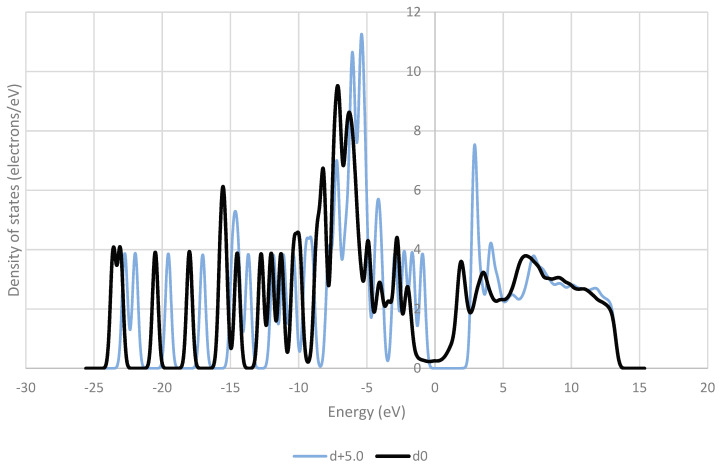
Comparison of the density of states (DOS) for paracetamol at the optimised distance from an aluminium (111) surface and at a distance of 5 Å away from the optimised position on the surface.

**Figure 12 pharmaceutics-16-00433-f012:**
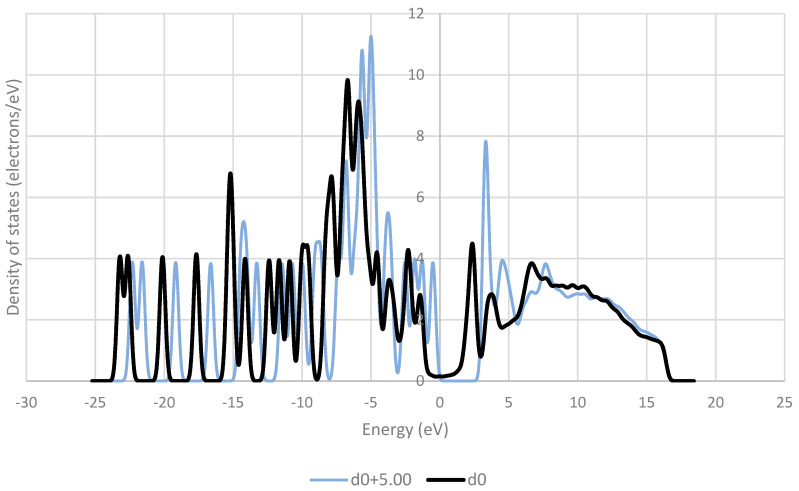
Comparison of the density of states (DOS) for paracetamol at the optimised distance from a copper (111) surface and at a distance of 5 Å away from the optimised position on the surface.

**Figure 13 pharmaceutics-16-00433-f013:**
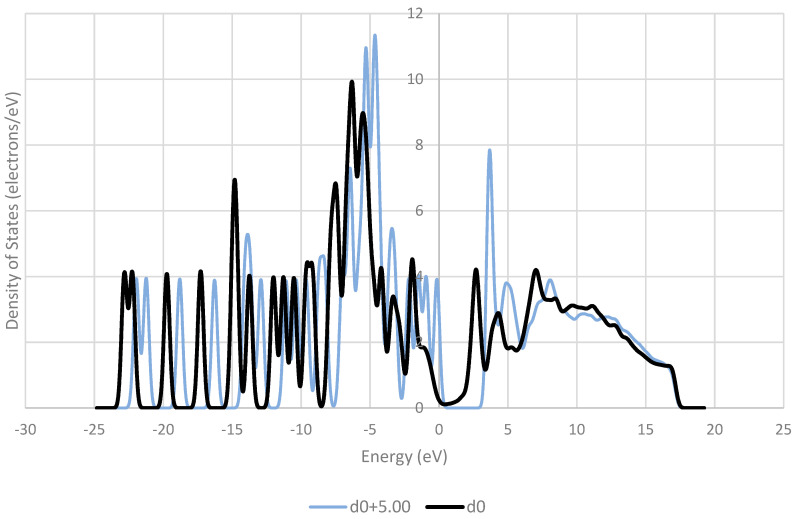
Comparison of the density of states (DOS) for paracetamol at the optimised distance from a nickel (111) surface and at a distance of 5 Å away from the optimised position on the surface.

**Figure 14 pharmaceutics-16-00433-f014:**
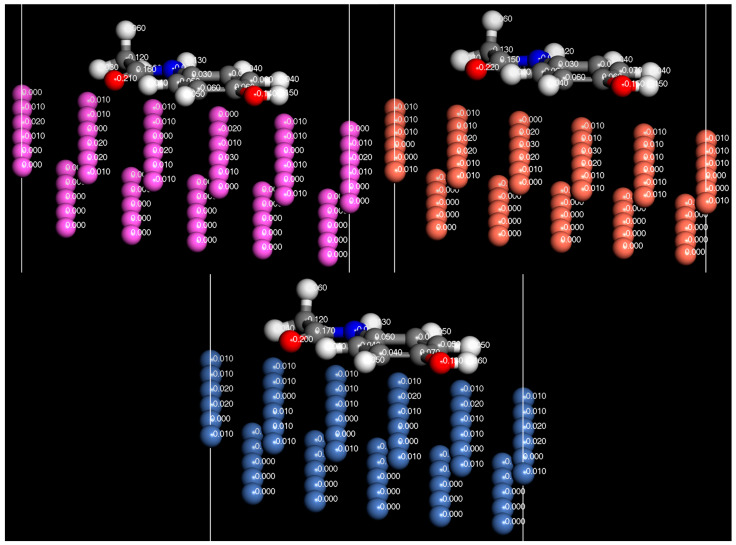
Hirshfeld charges mapped onto their respective atoms. Aluminium (**top left**). Copper (**top right**). Nickel (**bottom**). Side view.

**Figure 15 pharmaceutics-16-00433-f015:**
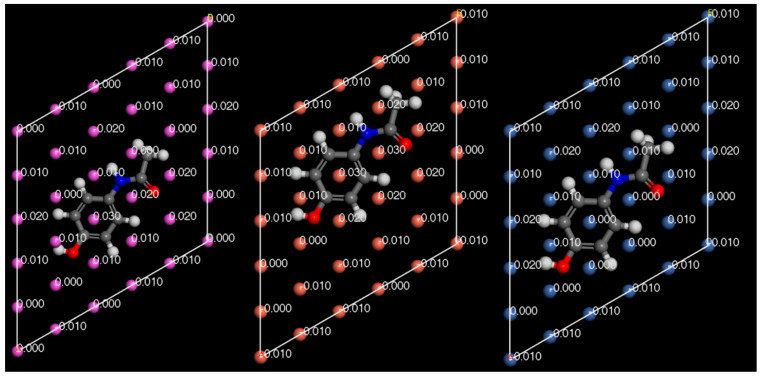
Surface Hirshfeld charges mapped on to their respective surface atoms. Aluminium (**left**). Copper (**middle**). Nickel (**right**). Bottom view.

**Figure 16 pharmaceutics-16-00433-f016:**
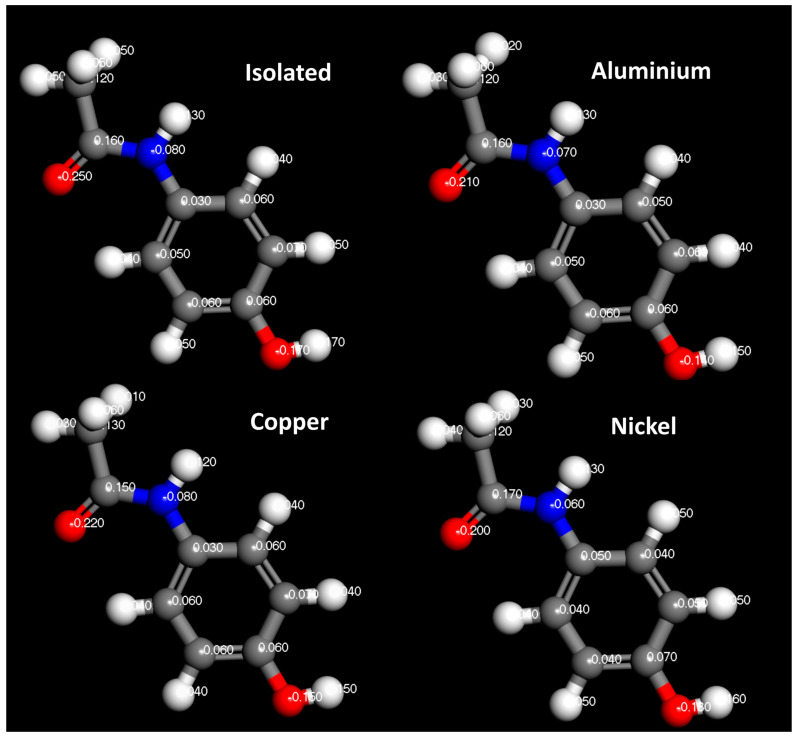
Hirshfeld charges of paracetamol molecule. Isolated (**top left**), on aluminium surface (**top right**), on copper surface (**bottom left**), on nickel surface (**bottom right**).

**Figure 17 pharmaceutics-16-00433-f017:**
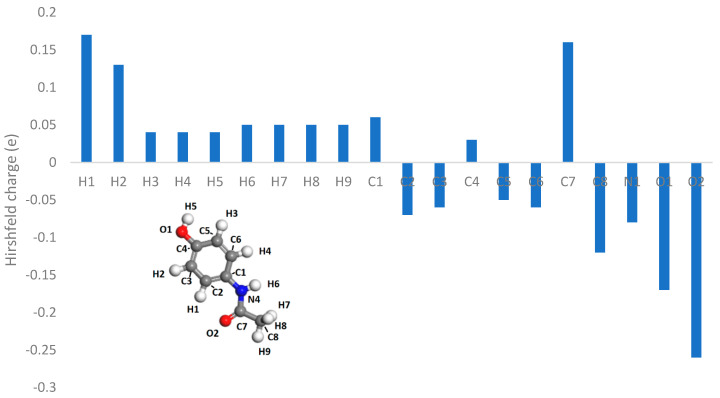
Hirshfeld charges on each atom of an isolated paracetamol molecule.

**Figure 18 pharmaceutics-16-00433-f018:**
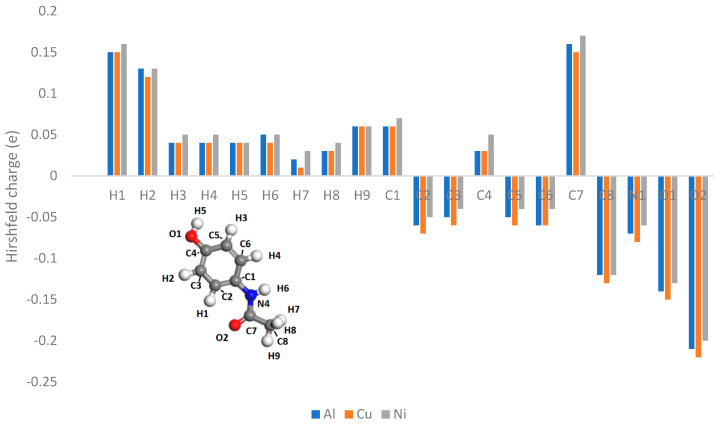
Hirshfeld charges on each atom of a paracetamol molecule placed on a (111) metal surface.

**Figure 19 pharmaceutics-16-00433-f019:**
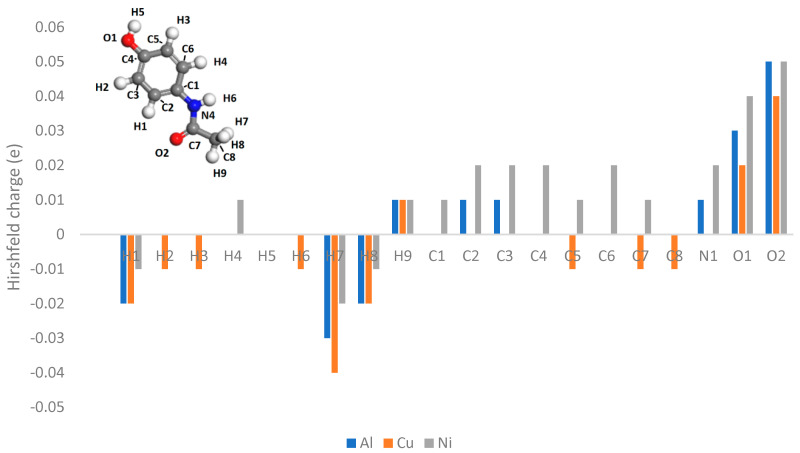
Difference between Hirshfeld charge at the surface and the Hirshfeld charge 5 Å from the surface.

**Table 1 pharmaceutics-16-00433-t001:** Kinetic energy cutoff, Monkhorst–Pack grid, and number of irreducible k-points used for each simulation during DFT and COMPASSIII comparison.

System—Xc Functional	Kinetic Energy Cutoff	MP Grid Sampling (*a* × *b* × *c*)	No. of k-Points
Al—LDA, CA-PZ	210	6 × 6 × 6	10
Al—GGA, PBE	210	6 × 6 × 6	10
Cu—LDA, CA-PZ	450	8 × 8 × 8	20
Cu—GGA, PBE	450	8 × 8 × 8	20
Ni—LDA, CA-PZ	440	8 × 8 × 8	20
Ni—GGA, PBE	440	8 × 8 × 8	20
Paracetamol—LDA, CA-PZ	630	4 × 3 × 2	8
Paracetamol—GGA, PBE	630	4 × 3 × 2	8

**Table 2 pharmaceutics-16-00433-t002:** Experimental and calculated lattice parameters of aluminium (Al), copper (Cu), nickel (Ni), and paracetamol (Paracet.) Unit cell lengths (*a*, *b*, *c*). Unit cell angles (α, β, γ). Lattice parameters of Al, Cu, and Ni were obtained at temperatures of 25 °C, 18 °C, and 25 °C, respectively, according to Wyckoff [29]. The lattice parameters of paracetamol were obtained at −192 °C, as reported by Bruhn et al. [30].

		Lattice Parameters (Å)					
		*a*	*b*	*c*	*α*	*β*	*γ*
Al	Experimental [29]	4.05	-	-	90	90	90
	DFT—(LDA, CA-PZ)	3.99	-	-	90	90	90
	DFT—(GGA, PBE)	4.04	-	-	90	90	90
	MM—COMPASSIII	4.04	-	-	90	90	90
Cu	Experimental [29]	3.60	-	-	90	90	90
	DFT—(LDA, CA-PZ)	3.52	-	-	90	90	90
	DFT—(GGA, PBE)	3.63	-	-	90	90	90
	MM—COMPASSIII	3.61	-	-	90	90	90
Ni	Experimental [29]	3.54	-	-	90	90	90
	DFT—(LDA, CA-PZ)	3.42	-	-	90	90	90
	DFT—(GGA, PBE)	3.51	-	-	90	90	90
	MM—COMPASSIII	3.52	-	-	90	90	90
Paracet.	Experimental [30]	7.07	9.19	11.49	90	98.64	90
	DFT—(GGA, PBE)	7.02	9.04	11.76	90	99.32	90
	MM—COMPASSIII	7.14	8.91	11.70	90	97.03	90

**Table 3 pharmaceutics-16-00433-t003:** DFT-predicted Fermi energy, vacuum energy for each isolated system, and calculated work function.

System	Fermi Energy (eV)	Vacuum Energy (eV)	Theoretical Work Function (eV)
Isolated Paracetamol	−1.673	0.012	1.685
Al Surface	−2.215	0.073	2.288
Cu Surface	−2.879	0.073	2.952
Ni Surface	−3.403	0.057	3.460

## Data Availability

The raw data supporting the conclusions of this article will be made available by the authors on request.

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
