# Peer review of "Triboelectric Charging Properties of the Functional Groups of Common Pharmaceutical Materials Using Density Functional Theory Calculations"

_pharmaceutics, 2024, doi:10.3390/pharmaceutics16030433_

Round 1
Reviewer 1 Report
Comments and Suggestions for Authors
The manuscript entitled “Triboelectric Charging Properties of the Functional Groups of Common Pharmaceutical Materials Using Density Functional Theory Calculations” presents results of the density functional theory calculations used to investigate the charge transfer characteristics of several functional groups of paracetamol on various metallic substrates, i.e., Al, Cu or Ni. The manuscript is well-prepared and it could be of the interest of a potential reader as it deals with a triboelectric phenomenon. I think the manuscript should be published in the Journal after minor amendments that could improve the quality of the manuscript.
1. The authors should explain why they have selected Al, Cu or Ni metals as the surfaces. Are these metals often in contact with paracetamol?
2. The authors should explain the distances from the surface they have selected to calculations.
3. Did the authors use a single paracetamol molecule in their calculations (with contact with the various metallic surfaces)?
4. Table 2: (i) Errors/uncertainties for the calculated values should be added. (ii) The temperature at which cell parameters were obtained should be added. (iii) The biggest/highest discrepancies between the experimental data and theoretical calculations are found for paracetamol. What is the reason? In the experiment, the sample is finite and there are some interactions between molecules (short-range and long-range, bonding, etc.). The boundary conditions, in some sense, imitate such conditions, etc. In addition, the molecules are not so perfect arranged in the realm world. I think the authors should explain these discrepancies between experimental data and theoretical calculations for paracetamol.
5. Table 3: The fifth column is empty.
6. Page 8, lines 248-258: Figure 5 is not cited in the main text.
Author Response
Reviewer comments (highlighted in blue)
My draft response (Black)
Thank you for your valuable feedback and constructive comments on our manuscript. We appreciate the time and effort you have dedicated to reviewing our work. Below, we have addressed each of your comments and suggestions. The changes in the manuscript are given in highlighted form below.
- The authors should explain why they have selected Al, Cu or Ni metals as the surfaces. Are these metals often in contact with paracetamol?
No, in fact they are not used at all! Stainless steel is the most common material, but as it is an alloy of various metals and carbon, it was decided not to use it. In order to get a fundamental understanding of the charge transfer processes, Al, Cu or Ni were used instead, for which the work function can be precisely calculated and measured. Both the experimentally and theoretically obtained work functions of aluminium and copper closely align with the work function for stainless steel, as indicated by (Kawano, 2022) and (D'Arrigo, 1978), respectively.
From a technical perspective, modelling stainless steel using first principles is challenging due to its alloy nature. This necessitates making several assumptions regarding the real chemical composition at the surface. In contrast, our study focuses on the well-defined (111) surfaces of aluminium, copper, and nickel. The selection was based on the similarity of their work function to that of stainless steel. Notably, these metals are all face-centred cubic (FCC) metals, resulting in comparable (111) surfaces. Moreover, they exhibit a range of work functions, reflecting distinct charging propensities. References provided below.
D’Arrigo, J. S. Screening of membrane surface charges by divalent cations: an atomic 630 representation. Am. J. Physiol. Physiol. 235, C109–C117 (1978).
Kawano, “Effective Work Functions of the Elements: Database, Most probable value, Previously recommended value, Polycrystalline thermionic contrast, Change at critical temperature, Anisotropic dependence sequence, Particle size dependence,” _Prog. Surf. Sci._, vol. 97, no. 1, p. 100583, 2022, doi: https://doi.org/10.1016/j.progsurf.2020.100583.
- The authors should explain the distances from the surface they have selected to calculations.
The selection of distances from the surface was informed by prior experience. It was expected that the most significant interactions would manifest in close proximity to the surface, rapidly tapering off as the distance increased. A maximum distance of 5Å (x_max) was considered suitable, marking a position at which negligible interactions between the surface and the molecule were anticipated. To optimize computational efficiency, the distances were halved successively (i.e., (x_max/2), (x_max/4), (x_max/8), (x_max/16), ...), ensuring a more concentrated sampling in the most interesting region near the surface. This method enhances the resolution in the region of greatest interest, but also significantly reduces the overall number of calculations required.
- Did the authors use a single paracetamol molecule in their calculations (with contact with the various metallic surfaces)?
CASTEP uses periodic cells as the input structure, which models systems as infinitely repeating unit cells in each direction. In the preliminary phases of this study, a convergence analysis was conducted on an isolated paracetamol. The results revealed that within cells of 10 volume, no interactions were observed between the paracetamol molecules. Consequently, the simulations were designed to represent a singular paracetamol molecule on a surface.
- Table 2: (i) Errors/uncertainties for the calculated values should be added. (ii) The temperature at which cell parameters were obtained should be added. (iii) The biggest/highest discrepancies between the experimental data and theoretical calculations are found for paracetamol. What is the reason? In the experiment, the sample is finite and there are some interactions between molecules (short-range and long-range, bonding, etc.). The boundary conditions, in some sense, imitate such conditions, etc. In addition, the molecules are not so perfect arranged in the realm world. I think the authors should explain these discrepancies between experimental data and theoretical calculations for paracetamol.
After careful consideration, we have decided not to include uncertainties in the presented results to maintain the readability and clarity of the tables. We have employed established computational methodologies and techniques that are widely accepted in the field. The experimentally obtained lattice parameters are provided as reported in the reference, to an appropriate number of significant figures. Additionally, our study focuses on the probing triboelectric charge transfer properties of paracetamol and is not focused on the prediction of lattice parameters. To be concise we have chosen to prioritize the presentation of key findings and their implications in the manuscript. Including details on errors and uncertainties, which do not significantly affect the main outcomes of the study, could potentially detrimentally impact the readability of the paper.
Added - All geometry optimisation and DFT calculations were performed at 0 K. It is added to text (Line 121). Experimentally obtained conditions added (Line 139).
As previously mentioned, an initial study was carried out to ensure that there were negligible long-range interactions between the paracetamols in the adjacent periodic cells. The role of temperature in changing work function and charge transfer is well known and is a topic that is beyond the scope of this work. However, we are interested in exploring this using first principles modelling to investigate this in our future research.
In this context, variance in lattice parameters between theoretical predictions and experimental observations stems from the intrinsic complexity of molecular crystals, like paracetamol, compared to pure metals. Such materials have multiple elements, several bonding types, and long-range interactions, necessitating a more rigorous modelling approach. For instance, to accurately capture long-range van der Waals forces, the incorporation of dispersion correction schemes is essential, representing a developing area of study in its own right. Moreover, the elevated complexity and fragility of molecular crystals also amplifies the challenges associated with experimental measurements. Additionally, variations in the experimental approaches utilized by different research groups inevitably contribute to minor discrepancies in reported values, particularly when compared with DFT calculations conducted at 0 K. The authors consider that even though small discrepancies are reported in this study the match between experimental and predicted lattice parameters is still very good and similar to that obtained for the metals.
- Table 3: The fifth column is empty.
Apologies for the formatting oversight. A fifth column was unintended and contains no information. This issue will be rectified during the typesetting stage by the publisher.
- Page 8, lines 248-258: Figure 5 is not cited in the main text.
Figures 5 is mentioned in the text on page 11, line 220. However, it mentions Figures 4 and 5 together.
It is hoped the responses given are satisfactory to the reviewer. Thank you once again for your thoughtful review and for helping us improve the quality of our work.
Reviewer 2 Report
Comments and Suggestions for Authors
The manuscript "Triboelectric charging properties of the functional groups of common pharmaceutical materials using density functional theory calculations." has a n important and actual subject of the specific research field.
The presented results are interesting and useful.
The specific comments are:
1. The paper is very well-organized. The article is written in a manner accessible to those working in the traditional fields of science and engineering.
2. The paper's title is brief, and it reflects the main theme of the paper.
3. The abstract concisely conveys the argument and conclusions of the manuscript.
4. The keywords are suitable so the article can be found in the current registers or indexes.
5. The introduction section includes a brief overview of the important research on the topic addressed.
6. In the Experimental section, the authors present the chemicals and materials used, the methods and procedures.
7. The measurement units are written according to the International Standards.
8. Results and Discussion Section: The authors present and interpret the results of the experiments performed. This section is very well organized.
9. Conclusion Section: The authors mentioned the major and specific conclusions of their research study.
10. Figures and tables: The tables and figures are numbered sequentially, and they are clearly labeled and positioned close to the relevant text. Titles of tables and figures are brief and informative. All the tables and figures included are referred.
11. The abbreviations and nomenclature are used according with applicable international standards and rules.
12. The length of the manuscript is adequate.
13. The references must write at MDPI standards.
14. The supplementary material completing the presented results.
Author Response
Thank you for your kind comments and feedback on our manuscript, we appreciate the time and effort you have dedicated to reviewing our work and have made the suggested modification to the referencing style. The Multidisciplinary Digital Publishing Institute (MDPI) style is now used throughout. We hope that these changes address your concerns. Your input has been reassuring in the quality of our work, and we sincerely appreciate your time and effort in reviewing our manuscript.
Reviewer 3 Report
Comments and Suggestions for Authors
This work contributes to assessing the contribution of DFT calculations for the rationalization of empirically observed phenomena in the pharmaceutical industry. Choosing paracetamol as API is a success, due to its wide therapeutic use and the circumstances that contribute to the moderate complexity of its chemical structure. The illustration of the work seems impeccable to me. And the Conclusions, presented with the clarity and rigor that the work provides.
Author Response
Thank you for your feedback and for expressing confidence in our work by suggesting no corrections. We appreciate the time and effort you have dedicated to reviewing our work and appreciate your positive evaluation and are grateful for your attention to our manuscript. Your confidence in our research is greatly valued.